# Monocular Depth Estimation: Lightweight Convolutional and Matrix Capsule Feature-Fusion Network

**DOI:** 10.3390/s22176344

**Published:** 2022-08-23

**Authors:** Yinchu Wang, Haijiang Zhu

**Affiliations:** College of Information Science & Technology, Beijing University of Chemical Technology, Beijing 100029, China

**Keywords:** depth estimation, convolutional neural network, matrix capsule feature, feature fusion

## Abstract

This paper reports a study that aims to solve the problem of the weak adaptability to angle transformation of current monocular depth estimation algorithms. These algorithms are based on convolutional neural networks (CNNs) but produce results lacking in estimation accuracy and robustness. The paper proposes a lightweight network based on convolution and capsule feature fusion (CNNapsule). First, the paper introduces a fusion block module that integrates CNN features and matrix capsule features to improve the adaptability of the network to perspective transformations. The fusion and deconvolution features are fused through skip connections to generate a depth image. In addition, the corresponding loss function is designed according to the long-tail distribution, gradient similarity, and structural similarity of the datasets. Finally, the results are compared with the methods applied to the NYU Depth V2 and KITTI datasets and show that our proposed method has better accuracy on the C1 and C2 indices and a better visual effect than traditional methods and deep learning methods without transfer learning. The number of trainable parameters required by this method is 65% lower than that required by methods presented in the literature. The generalization of this method is verified via the comparative testing of the data collected from the internet and mobile phones.

## 1. Introduction

Depth estimation is a branch of basic computer-vision research. It depends on professional depth acquisition equipment and robust computer-vision algorithms. Obtaining a depth image of a real-world scene through depth estimation provides data that can serve as the basis for many applications, such as robots [1], autonomous driving [2], SLAM [3], augmented reality [4], 3D reconstruction [5], and segmentation [6]. There are many depth estimation methods, including structured light [7], time of flight (TOF) [8], binocular vision [9], and monocular vision [10]. The structured light and TOF methods are highly accurate indoors, but they are easily affected by scattered light and multiple reflections outdoors, distorting measurement data. Although binocular vision is applicable both indoors and outdoors, the camera baseline limits the measurement range, and the measurement effect of weak-texture areas in the scene is poor. Compared with other methods, the monocular vision method can be applied to indoor and outdoor scenes and has a wide field of vision, simple structure, and low cost. Therefore, few studies have recently focused on monocular vision [11,12,13,14,15,16].

Early monocular depth estimation methods utilized traditional computer-vision methods [17,18]. With the rapid development of GPUs, monocular depth estimation based on a CNN has become common. In 2014, Eigen [19] adopted the CNN structure in monocular depth estimation, achieving better accuracy than that of traditional methods, but the resolution of the generated depth image was low, only 80 × 60 pixels. Laina [20] proposed a complete convolutional depth estimation network with an encoder–decoder structure to improve image resolution using an up-sampling method and treating the first inverse Huber loss as the optimization function. Moreover, this method achieved better estimation accuracy by increasing the number of network layers. However, they still did not fully use multi-scale information, which limited the further improvement of the estimation accuracy. To solve this problem, Xu [21] proposed a depth estimation network based on a conditional random field model in 2017 by extracting multi-scale feature maps and fused features and achieving superior results. However, the estimation accuracy was still not high because the context information was not used in network reasoning. Hao [22] adopted ResNet-101 as the backbone network to take full advantage of context information based on transfer learning research. They exploited hole convolution to extract texture features and constructed an attention fuse block; the channel reduced block-to-fuse features in the decoding stage with high accuracy. Since then, many studies on monocular depth estimation have taken advantage of transfer learning and have used different backbone networks to improve the depth estimation accuracy, such as the VGG model [23] and the DenseNet model [24,25]. However, the CNN still cannot consider perspective transformation. To reduce the impact of perspective transformation on model generalization performance, these methods need to design overly complex network structures to adapt to the different perspective images, which results in a large number of parameters and limited generalization performance. To overcome these shortcomings, Hinton [26] proposed a robust capsule network. The capsule network constructed a 16-dimensional vector as a capsule to characterize object features and achieved better performance than baseline CNN. Further, Hinton [27] transformed the vector features into matrix capsule features with the GMM (Gaussian Mixture Model). Further, the matrix capsule features had good perspective transformation ability. Meanwhile, they used the EM algorithm to extract features and realized end-to-end training. After that, many scholars have made improvements on these two original networks. Ribeiro [28] proposed a better capsule routing algorithm derived from Variational Bayes for fitting a mixture of transformation Gaussians. Gu [29] proposed interpretable GraCapsNets (Graph Capsule Networks) and replaced the routing part with a multi-head attention-based graph pooling approach. Ribeiro [30] proposed an alternative global view based on representing the inherent uncertainty in part-object assignment and accelerate the network without sacrificing performance. Sabour [31] proposed a way to learn primary capsule encoders that detect atomic parts from a single image and improve the adaptability of occlusion and cluttered backgrounds. Ribeiro [32] introduced many capsule networks and explored the extensive applications of capsule networks in different fields. The above methods retain perspective transformation based on matrix capsule features. However, there are few studies on the application of capsule networks in monocular depth estimation. Moreover, whether the combination of capsule networks and CNNs has better performance has not been fully verified. Based on the above considerations, this study developed a lightweight monocular depth estimation network (CNNapsule) that integrates the matrix capsule feature into the CNN model. On the basis of reducing the number of parameters and improving generalization, this paper verified the effectiveness of the combination of capsule networks and CNNs in performing monocular depth estimation. The main contributions of this paper are as follows:A fusion block that can simultaneously obtain the matrix capsule feature and CNN feature of the same scene;A method for generating depth images by integrating the three-feature information in the encoder stage, decoder stage, and fusion block;A triple loss function is designed with depth difference, gradient difference, and structural similarity.

## 2. Convolutional Capsule Feature-Fusion Network (CNNapsule)

### 2.1. CNNapsule Network

The CNNapsule presented in this paper adopts the encoder–decoder structure, as shown in Figure 1. When the size of the input images is 256 × 256 × 3 pixels, the problem of high computational complexity arises. Therefore, the step size is set to 2, and 5 × 5 convolution blocks are first utilized to extract the shallow features in our CNNapsule network. Then, four feature-extraction modules extract multi-scale features. Each block includes one convolution layer, one batch normalization (BN) layer, one ReLU activation function, one dropout layer (rate = 0.2), and an average pooling layer. Simultaneously, the feature of the matrix capsule is extracted according to the third module. The fusion features consistent with the size of the CNN feature map can be obtained through the fusion block. In the decoder stage, the fusion features (yellow blocks), CNN features (orange blocks), and decoder features (blue blocks) are spliced together through a skip connection. The spliced features are input to the next layer. For higher-resolution images, the network adopts the deconvolution operation and the Leaky ReLU activation function (α = 0.2). Finally, a depth image with 128 × 128 pixels is obtained through one convolution with a 1 × 1 pixel size.

### 2.2. Matrix Capsule Feature Description

In current CNN methods for feature representation, the learning of perspective transformation mostly depends on data enhancements such as translation, flip, and rotation. These methods resist attacks poorly; thus, Hinton [26] proposed the CapsNet network in 2017 and improved feature representation. This method achieved high accuracy on the MNIST and Cifar10 datasets by replacing CNN’s information transmission unit neuron with a capsule. Then, they transformed the vector representing the feature into a pose matrix in CapsNet and proposed a matrix capsule network [27], as shown in Figure 2, where N capsules characterize n targets. It was verified that the network had a good perspective transformation learning ability on the SmallNORB dataset.

Matrix capsule features are illustrated in Figure 3. The capsule set in layer l is represented by Il. Each capsule contains a post matrix M4×4 and an activation probability a1×1. For the i-th capsule of layer l and the j-th capsule of layer l+1, there is a trainable 4×4 conversion matrix Wij. Parameters Wij and two learnable offsets per capsule are the only stored parameters. The pose matrix of the i-th capsule is transformed by Wij, and the pose matrix of the j-th capsule is voted by Vij=MiWij. For all iϵIi and jϵIj, Vij and ai are taken as inputs. All pose matrices and activation probabilities of layer l+1 can be calculated using the EM algorithm. The solution steps of the EM algorithm are as follows (Algorithm 1):
**Algorithm 1.** EM algorithm**Procedure** EM algorithm returns activation and pose of the capsules in layer *l* + 1 based on activations and poses of capsules in layer *l*. Vijh is the hth dimension of the vote from capsule *i* with activation ai in layer *l* to capsule *j* in layer *L* + 1. mjh is the *h^th^* dimension of the pose from capsule *j*. Rij is initialized to 1Il+1.**M**-STEP for one higher-level capsuleRij=aiRij, ∀i∈Ilmjh=∑iRijVijh∑iRij, ∀hσjh2=∑iRijVijh∑iRij, ∀h                costh=γu+logσjh∑iRij               aj=logisticργv−∑hcosth**E**-STEP for one lower-level capsulepj=1∏h2πσjh2e−∑hVijh−mjh22σjh2, ∀j∈Il+1Rij=ajpj∑n∈Il+1anpn, ∀j∈Il+1

The network accepts all poses obtained from the last layer as the features of the matrix capsule. In the experiment, the number of iterations of the EM algorithm between capsules is set to three to obtain good matrix capsule characteristics.

### 2.3. Fusion Block

Most depth estimation networks are based on CNNs, and the CNN model’s lack of perspective transformation ability also affects the accuracy of depth estimation. Therefore, we designed a network structure by introducing the matrix capsule feature. The fusion block (FB) structure easily provides feature fusion, as shown in Figure 4. First, the matrix capsule feature is reshaped into one vector with a length of n+1. Next, the n+1 vector is mapped onto another vector with a length of $n^{2}$ using a fully connected layer. Then, the resulting vector is remapped to an n×n characteristic graph. To further improve the diversity of feature maps, m kinds of feature maps of size n×n are generated through a matrix capsule network. They are spliced into an n×n×m characteristic graph. Finally, the network adopts three 1×1 convolution layers to obtain the fusion features. The number of channels of the feature map is expanded to balance the contribution ratio of the matrix capsule and CNN features. In the experiment, n was set to 16, and m was set to 4 in the network parameters.

### 2.4. Loss Function

An effective loss function is often conducive to network training, speeding up the training speed and improving the overall depth estimation performance. Many kinds of loss-function designs optimize the network in the reported literature [19,25,33,34]. Jiao [33] found a long-tail distribution of depth values in the NYU depth V2 and KITTI datasets. This means that the contribution of hard examples with large depth values to model training is minimal, making the model more inclined to predict small depth values. To increase the contribution of hard examples to model training, we designed an adaptive depth loss function:(1)$$L_{depth}=\frac{1}{N}\sum_{p}^{N}y_{p}^{*}\left[f\left(y_{p}^{*}\right)-f\left(y_{p}\right)\right]$$
(2)$$f\left(y\right)=\frac{maxDis}{y}$$
where N is the number of pixels in the image, $y_{p}^{*}$ is the true value of the depth map, $y_{p}$ is the predicted depth value of the depth estimation network, and maxDis is the maximum depth value in the dataset. In the comparison experiment, m was set to 1000.0 for the NYU depth V2 dataset, with an effective depth range of 0–10 m, and to 8000.0 for the KITTI dataset, with an effective depth range of 0–80 m.

To make full use of the edge information of the depth map, this paper utilized the gradient similarity to construct gradient loss $L_{grad}$:(3)$$L_{grad}=\frac{1}{N}\sum_{p}^{N}\left|g_{x}\left[f\left(y_{p}\right),f\left(y_{p}^{*}\right)\right]\right|+\left|g_{y}\left[f\left(y_{p}\right),f\left(y_{p}^{*}\right)\right]\right|$$
where $g_{x}$ and $g_{y}$ represent the calculated gradients of $f\left(y\right)$ and $f\left(y^{*}\right)$ in the x and y directions, respectively.

Considering the influence of structural similarity (*SSIM*) on depth estimation, structural similarity loss $L_{SSIM}$ is calculated as follows:(4)$$L_{SSIM}=1-SSIM\left[f\left(y_{p}\right),f\left(y_{p}^{*}\right)\right]$$

Taken together, the overall loss function $L_{cost}$ in this paper is given by:(5)$$L_{cost}=L_{depth}+L_{grad}+\lambda L_{SSIM}$$

In this experiment, λ was set to 0.5.

## 3. Evaluation Indicators

In this paper, the proposed method is quantitatively compared with existing methods according to the seven evaluation indices [19] proposed in the previous study. These evaluation indicators are named the C1 index and C2 index.

The C1 indices are the average relative error (*AbsRel*), the root mean square error (*RMSE*), the log mean error (*log*10), and the log root mean square error (*logRMS*), which are given by: (6)$$AbsRel=\frac{1}{N}\sum_{p}^{N}\frac{\left|y_{p}-y_{p}^{*}\right|}{y_{p}^{*}}$$
(7)$$RMSE=\sqrt{\frac{1}{N}\sum_{p}^{N}{\left(y_{p}-y_{p}^{*}\right)}^{2}}$$
(8)$$\log\left(10\right)=\frac{1}{N}\sum_{p}^{N}\left|log_{10}\left(y_{p}\right)-log_{10}\left(y_{p}^{*}\right)\right|,\;\mathrm{and}$$
(9)$$\log\left(RMS\right)=\frac{1}{N}\sum_{p}^{N}{\left|log_{10}\left(y_{p}\right)-log_{10}\left(y_{p}^{*}\right)\right|}^{2}$$

The threshold accuracy $\left(\delta_{i}\right)$ is the percentage of the number of pixels satisfying Equation (10) in the predicted value $y_{p}$ and the real value $y_{p}^{*}$ in the number of pixels of the input image:(10)$$max\left(\frac{y_{p}}{y_{p}^{*}},\frac{y_{p}^{*}}{y_{p}}\right)=\delta <\delta_{i}$$
where three thresholds in $\delta_{i}=1.25^{i}$ are used for quantitative comparison, that is, $\delta_{1}=1.25$, $\delta_{2}=1.25^{2},$ and $\delta_{3}=1.25^{3}$. We call the three thresholds the C2 indices.

To prevent the influence of missing values on error calculation, the part with missing values in the real depth image is filtered. In addition, the seven indices are calculated when the pixels are present only in the real depth images.

## 4. Experimental Results and Analysis

First, we present the data-expansion method. The quantitative analysis using the C1 and C2 indices was carried out on the NYU depth V2 [35] and KITTI datasets [36], and the qualitative analysis was compared with the reported methods. The qualitative comparison experiment was conducted to better verify the generalization performance of the constructed network by selecting some real indoor images (for NYU depth V2), and outdoor road images (for KITTI) collected from the internet and reality. The experiment used the Tensorflow deep learning framework to build the network, and the processor was Intel (R) core (TM) i7-10750h CPU @ 2.6 GHz. The graphics card was an NVIDIA Geforce RTX 2060. The Tensorflow version was 1.13.0; the CUDA version adopted was cuda10.0.

### 4.1. Data Augmentation

The data-augmentation strategy employed in many papers is an important means of improving the generalization performance of a deep network. This experiment used a random online transformation to expand the training data. The expansion methods were as follows:Brightness: The input image’s brightness was changed with a probability of 0.5, in the brightness range of [0.5, 1.5];Contrast: The contrast of the input image was 0.5, with a probability of changing the contrast to 0.5;Saturation: The input image’s saturation was changed with a probability of 0.5. The saturation range was [0.4, 1.2];Color: The R and G channels of the input image were exchanged with a probability of 0.25;Flip: The input and depth images were flipped horizontally with a probability of 0.5;Pan: The input image was randomly cropped to 224 × 224. To adapt to the network structure, the input image was scaled to 256 × 256, and the true depth map was scaled to 128 × 128.

Part of the augmented results from the NYU depth V2 and KITTI datasets are shown in Figure 5. In this study, changes in the lighting of the actual scene were simulated by changing the input image’s brightness, contrast, and saturation. In addition, some studies proved the effectiveness of exchanging the R channel and G channel [25]. Due to the scaling in the translation operation adopted, the world space geometry of the scene could not be retained. To solve this problem, the depth value was divided by the scaling multiple to correct it (increase the image by s times to make the camera closer by s times) [19]. The horizontal flip used in this paper preserved the geometry.

### 4.2. Experiments on NYU Depth V2

The NYU depth V2 dataset was used for the quantitative comparative analysis of indoor scene depth estimation, including 120,000 training samples and 654 pairs of test samples. In the experiment, the input RGB image was from 640 × 480 down to 256 × 256, and the true depth image was from 640 × 480 down to 128 × 128. In addition, there was no additional filling for the points without depth information in the real depth image in the experiment. The initial learning rate of each layer in the network was 0.01, and the momentum was set to 0.9. Due to the limitation of VRAM of GPU, the maximum batch size could only be set to four, and the occupation of VRAM was 4.8 G. The total training took 52 h and 750,000 iterations. To more intuitively illustrate the depth estimation effects of different methods, the depth estimation results of different methods were scaled to the same size.

During the test, the resolution of the predicted image was adjusted to 640 × 480 pixels using bilinear interpolation. The quantitative evaluation experiment adopted the predefined center-clipping method proposed by Eigen [19]. Since the constructed network does not use the “pre-training + fine-tuning” training mode, this method was quantitatively compared with the traditional method and deep learning algorithm without the pre-training model. The comparison of the C1 and C2 indices of different methods on NYU Depth V2 is shown in Table 1, where the evaluation index used the C1 and C2 indices, FB indicates whether the fusion block module is adopted, “×” stands for the unused module, “√” represents the use of the model, and bs indicates the batch size set in training. Compared with other methods, except for Zhou’s, five of the six evaluation indices were the best in this method (bs = 4, fusion block, $L_{cost}$), and the network output a higher resolution depth estimation image. Moreover, the errors between our method and Zhou’s method were also small.

To intuitively show the quality of the generated depth image, this section presents a qualitative comparison of different methods. Pseudo-color processing was performed on the generated depth estimation image to obtain a better visual effect. These results are shown in Figure 6. We can see that the proposed method obtained clear contour information for the bookshelf and generated better texture information for the table lamp. These results verified that the depth image generated using our method was of superior quality.

This paper also presents a qualitative comparison between the proposed method and the network model after transfer learning. The results are shown in Figure 7. Although the methods discussed in this paper were not pre-trained, the depth estimation results still had similar visual effects when using these methods and even clearer texture information for some samples. For example, due to the long-tail distribution effect on the dataset, there were more accurate estimation results in areas with large depth in the first and second lines. In addition, the comparison between the network parameters and the number of training iterations is shown in Table 2. The method described in this paper reduces the number of parameters by 65% and needs fewer iterations to converge.

### 4.3. Experiments in KITTI

The KITTI dataset consists entirely of stereo images of outdoor scenes captured by equipment installed on mobile vehicles. A 3D laser scanned the images. Images with a depth range of 0–80 m are often used for quantitative comparison and analysis of the depth estimation of roads and outdoor scenes. The RGB image resolution in the dataset is 1241 × 376 pixels, but the corresponding depth image has only a small density of depth information, and there are many pixels without depth information. According to the division method by Eigen [19], about 26,000 left-view images and corresponding depth images were extracted for training, and 697 test images were used for quantitative comparisons and analyses. For the missing depth information, this paper used the repair method provided in the KITTI toolkit to fill in the depth information. Similar to the preprocessing method used on NYU depth V2, the RGB image was sampled down to 256 × 256 pixels as the network input, and the real depth map was sampled down to 128 × 128 pixels as the network output and loss-function optimization. The hyperparametric design in the network was the same as in NYU depth v2. The training took 23 h and 325,000 iterations. The occupation of VRAM was 4.8 G.

The quantitative comparison results using our method and traditional deep learning methods without transfer learning on the KITTI dataset are shown in Table 3. It can be seen that our method achieved the lowest error under the three indicators. Even if the method by Zheng [15] limited the depth range to 1–50 m, our method still achieved a smaller estimation error. In addition, from the qualitative comparison with the method by Eigen [19], as shown in Figure 8, the edge of the depth image generated using our method was clearer, and the estimated depth information was more accurate, such as the automobile part in column 1 and column 3. These results also verified the effectiveness of adding gradient similarity and structural similarity to the design of the loss function.

### 4.4. Experiments on Collected Images

To further verify the generalization performance of the proposed method on indoor datasets, we collected indoor images from the internet and captured indoor images with mobile phones. The model trained on the NYU depth V2 dataset was directly used to test the collected samples. Some estimation results are shown in Figure 9. Even if the collection scenarios and collection methods of test and training samples were different, this method still obtained robust estimation results.

To verify the generalization performance of this algorithm on outdoor data, outdoor road data samples were collected from the network, directly estimating the monocular depth of the algorithm model trained on the KITTI dataset on these images. The estimation results are shown in Figure 10. Our method could still obtain certain scene texture information and was robust in estimating the relative depth of the scene. For example, the relative depths of two motorcycle riders and two vehicles provided good estimation results in the depth estimation results in row 4, which also verified the generalization performance of our method’s in-depth estimation on outdoor road data.

## 5. Conclusions

This paper describes a lightweight monocular depth estimation network based on convolution and capsule feature fusion. The basic architecture of the network adopts the encoder–decoder structure. The network integrates the matrix capsule feature and the CNN feature by constructing a fusion block to improve the adaptability of the network to perspective transformation. Moreover, considering the long-tail distribution effect of the dataset, an adaptive depth loss function is designed, and gradient similarity and structural similarity are introduced into the design of the loss function simultaneously. The experimental results show that the proposed method is better than the traditional method and the method without transfer learning and also has significant advantages in the number of parameters and similar visual effects with respect to methods using transfer learning. The generalization performance of this method is shown to be further proved using the collected images. On the other hand, there are still few research studies on the fusion method and adaptability with the routing algorithm. Therefore, how to further optimize the integration mode of our model is a challenge. Meanwhile, how to improve the routing algorithm to make it fit better with the fusion mode is another challenge. We leave this for our future work.

## Figures and Tables

**Figure 1 sensors-22-06344-f001:**
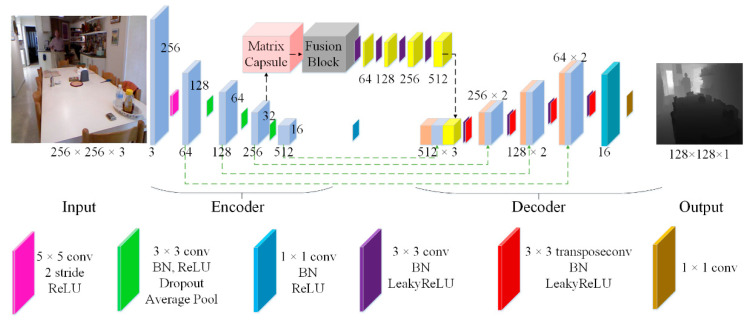
Architecture of the CNNaspule network.

**Figure 2 sensors-22-06344-f002:**
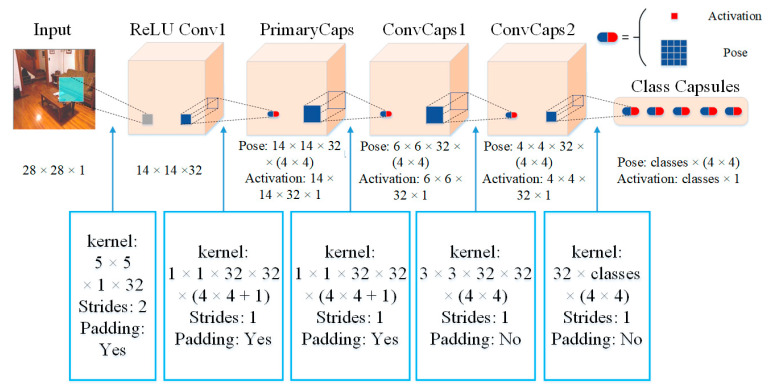
Architecture of the matrix capsule network.

**Figure 3 sensors-22-06344-f003:**
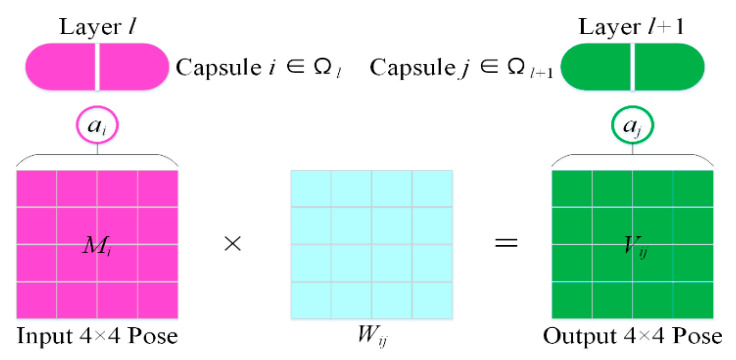
Illustration of the matrix capsule features with the pose matrix.

**Figure 4 sensors-22-06344-f004:**
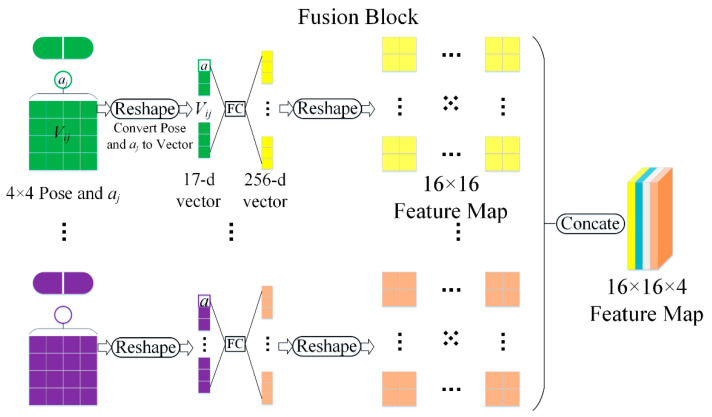
Illustration of converting matrix capsule features to feature maps.

**Figure 5 sensors-22-06344-f005:**
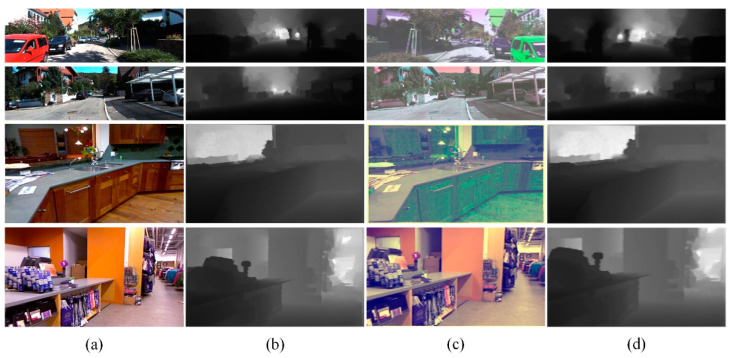
Partial data-augmentation effect on NYU depth V2 and KITTI datasets. (**a**) Original image; (**b**) ground truth; (**c**) data-augmentation results; (**d**) ground truth after data augmentation.

**Figure 6 sensors-22-06344-f006:**
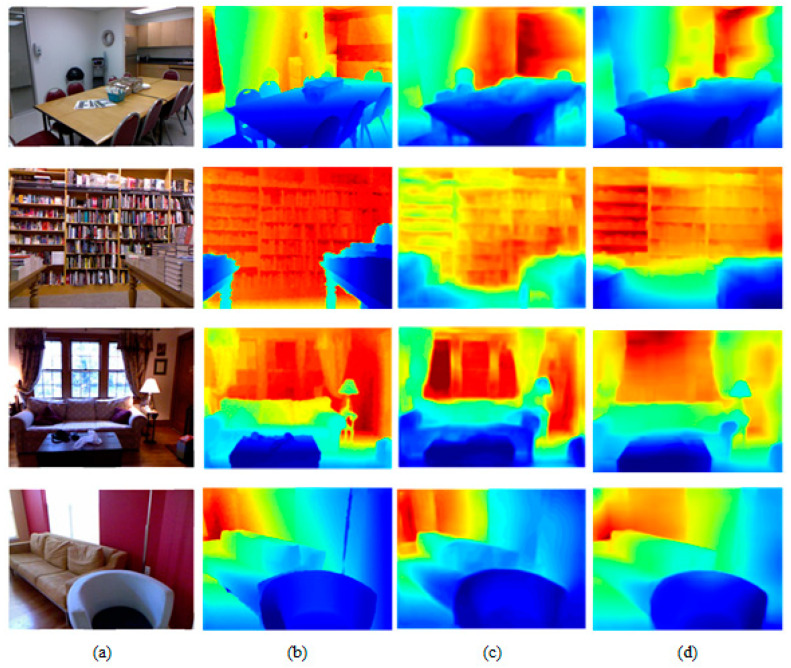
Comparison of monocular depth estimation results of NYU Depth V2 dataset. (**a**) RGB images; (**b**) ground truth; (**c**) Eigen’s method; (**d**) proposed method.

**Figure 7 sensors-22-06344-f007:**
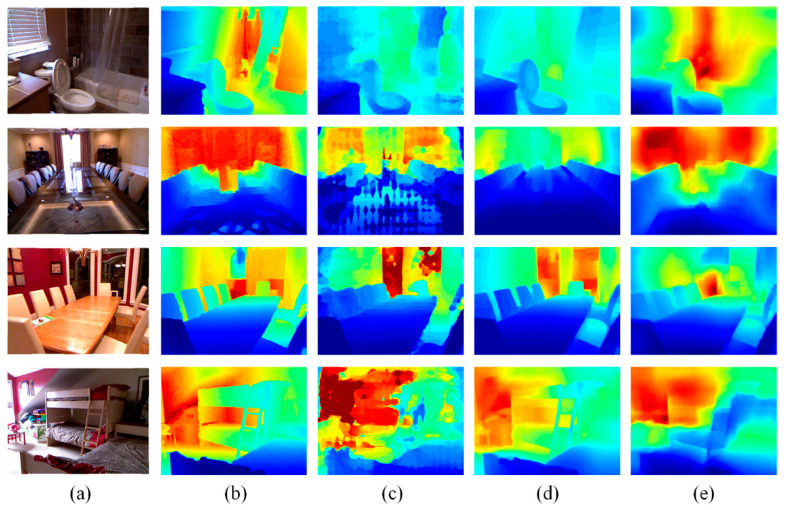
Visual effect comparison of the proposed method with transfer learning methods. (**a**) RGB images; (**b**) ground truth; (**c**) Fu’s method; (**d**) Alhashim’s method; (**e**) proposed method.

**Figure 8 sensors-22-06344-f008:**
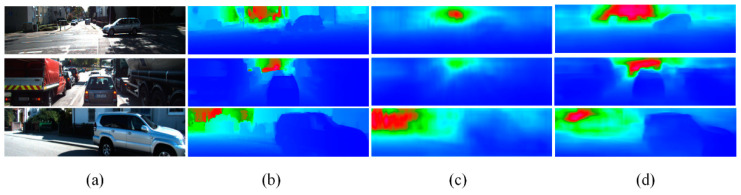
Comparison of monocular depth estimation results of KITTI dataset. (**a**) RGB images; (**b**) ground truth; (**c**) Eigen’s method; (**d**) proposed method.

**Figure 9 sensors-22-06344-f009:**
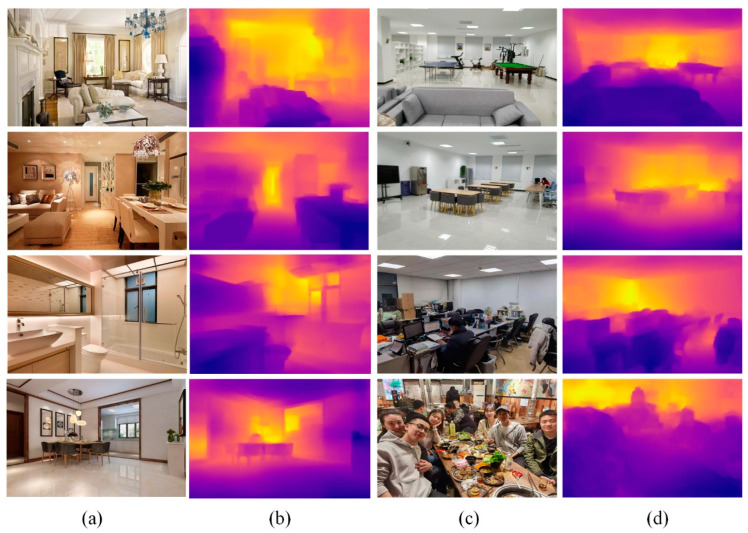
The results of the proposed method on the collected samples. (**a**) Images from the internet; (**b**) proposed method; (**c**) images captured with a mobile phone; (**d**) proposed method.

**Figure 10 sensors-22-06344-f010:**
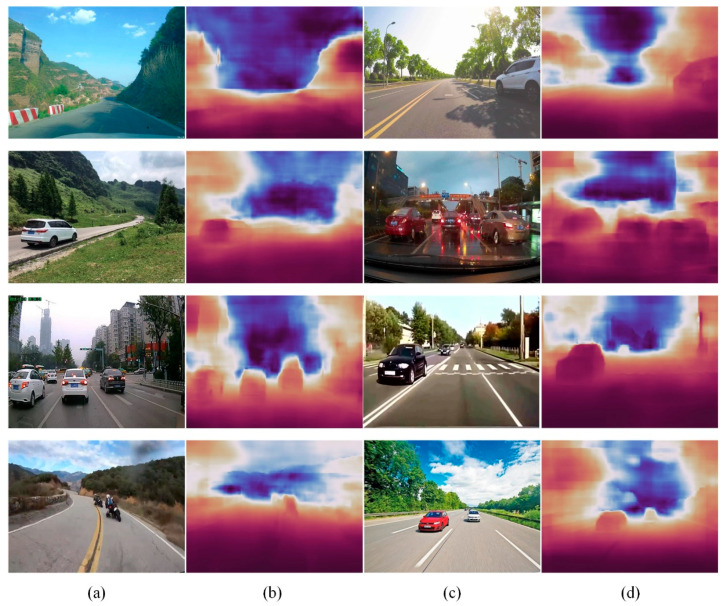
Depth estimation results of the proposed method on the collected road samples. (**a**,**c**) Images from the internet; (**b**,**d**) proposed method.

**Table 1 sensors-22-06344-t001:** Comparison of C1 and C2 indices of different methods on NYU Depth V2.

Method	FB	bs	Loss	C1 Indices			C2 Indices			Output
				AbsRel↓	RMSE↓	log10	δ1↑	δ2↑	δ3↑	
Zheng [15]	×	6	-	0.257	0.915	0.305	0.540	0.832	0.948	256 × 192
Liu [37]	×	-	-	0.230	0.824	0.095	0.614	0.883	0.971	-
Wang [38]	×	-	-	0.220	0.745	0.094	0.605	0.890	0.970	-
Zhou [39]	×	×	-	0.208	0.712	0.086	0.674	0.900	0.968	-
Lin [40]	×	×	-	0.279	0.942	-	0.501	-	-	-
Eigen [19]	×	32		0.215	0.907	-	0.637	0.887	0.971	80 × 60
Ours	×	4	Lcost	0.226	0.792	0.092	0.637	0.887	0.970	128 × 128
Ours	√	2	Lcost	0.216	0.757	0.088	0.657	0.897	0.973	128 × 128
Ours	√	4	Ldepth	0.229	0.817	0.094	0.605	0.883	0.971	128 × 128
Ours	√	4	Lcost	0.214	0.760	0.087	0.663	0.900	0.973	128 × 128

**Table 2 sensors-22-06344-t002:** Comparison of parameters and iterations between the proposed method and transfer learning methods.

Method	Parameters	Iterations
Fu [23]	110 M	3 M
Alhashim [25]	42.6 M	1 M
Ours	14.9 M	0.75 M

**Table 3 sensors-22-06344-t003:** Error comparison of different methods on the KITTI dataset.

Method	Depth Range	C1 Index		
		AbsRel↓	RMSE↓	logRMS↓
Make3D [41]	0–80 m	0.280	8.734	0.361
Eigen [19]	0–80 m	0.190	7.156	0.270
Zheng [15]	1–50 m	0.168	4.674	0.243
Ours (FB + Lcost, bs = 4)	0–80 m	0.163	3.873	0.226

## Data Availability

Not applicable.

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
