# Peer review of "Monocular Depth Estimation: Lightweight Convolutional and Matrix Capsule Feature-Fusion Network"

_sensors, 2022, doi:10.3390/s22176344_

Round 1

Reviewer 1 Report

Thanks for your article. There is merit to the study presented, but there a three main issues that need to be resolved.

Firstly, your interpretation of capsules and references used are a bit obsolete - including EM matrix capsules; there have been improved models since, e.g. Capsule routing via variational bayes, AAAI 2020

Perhaps this survery might be useful to update you on CapsNets:

Ribeiro, F.D.S., Duarte, K., Everett, M., Leontidis, G. and Shah, M., 2022. Learning with Capsules: A Survey. arXiv preprint arXiv:2206.02664.

and some others:

S. Sabour, A. Tagliasacchi, S. Yazdani, G. Hinton, and D. J. Fleet, “Unsupervised part representation by flow capsules,” in International Conference on Machine Learning. PMLR, 2021, pp. 9213–9223

F. De Sousa Ribeiro, G. Leontidis, and S. Kollias, “Introducing routing uncertainty in capsule networks,” in Advances in Neural Information Processing Systems, vol. 33, 2020, pp. 6490–6502.

J. Gu and V. Tresp, “Interpretable graph capsule networks for object recognition,” in Proceedings of the AAAI Conference on Artificial Intelligence (AAAI), 2020.

Secondly, your initial figures 1,2,4 are of low quality - maybe need updating; and your Table 1, the results involve studies from 5, 6, 8 years ago; that needs updating too with more recent methods.

Thirdly, that links to first point, EM routing might not be the best method to fuse the CNN with, but I think you could get away with if you frame it accordingly in the paper, that includes extensively expanding upon your capsule list of references in the paper

Reviewer 2 Report

Point 1 :     Please add approximate training time and VRAM share of GPU to 4.2 Experiments on NYU Depth V2 and 4.3 Experiments in KITTI.

Point 2 :      Correct the typo in the word CNNasrule on line 68.

Point 3 :    Please add a description of the EM algorithm in line 117.

Point 4 :    Please fill in the blanks such as bs, Loss, and Output in Table 1. If  you cannot fill in the blanks, please state the reason.

Round 2

Reviewer 1 Report

The authors made comprehensible changes to the paper which has improved extensively. The paper is ready for acceptance in its current form.